# 2.7 Å cryo-EM structure of ex vivo RML prion fibrils

Szymon W. Manka [1], Wenjuan Zhang [1], Adam Wenborn[1], Jemma Betts[1], Susan Joiner[1], Helen R. Saibil [2✉], John Collinge [1✉] & Jonathan D. F. Wadsworth [1✉]

Mammalian prions propagate as distinct strains and are composed of multichain assemblies of misfolded host-encoded prion protein (PrP). Here, we present a near-atomic resolution cryo-EM structure of PrP fibrils present in highly infectious prion rod preparations isolated from the brains of RML prion-infected mice. We found that prion rods comprise single-protofilament helical amyloid fibrils that coexist with twisted pairs of the same protofilaments. Each rung of the protofilament is formed by a single PrP monomer with the ordered core comprising PrP residues 94–225, which folds to create two asymmetric lobes with the N-linked glycans and the glycosylphosphatidylinositol anchor projecting from the C-terminal lobe. The overall architecture is comparable to that of recently reported PrP fibrils isolated from the brain of hamsters infected with the 263K prion strain. However, there are marked conformational variations that could result from differences in PrP sequence and/or represent distinguishing features of the distinct prion strains.

[1] MRC Prion Unit at UCL, Institute of Prion Diseases, University College London, 33 Cleveland Street, London W1W 7FF, UK. [2] Institute of Structural and Molecular Biology, Department of Biological Sciences, Birkbeck College, University of London, Malet Street, London WC1E 7HX, UK. ✉email: h.saibil@mail.cryst.bbk.ac.uk; jc@prion.ucl.ac.uk; j.wadsworth@prion.ucl.ac.uk

Prions are lethal infectious agents that cause fatal neurodegenerative diseases in mammals, including scrapie in sheep and goats, chronic wasting disease in cervids, bovine spongiform encephalopathy (BSE) in cattle and Creutzfeldt-Jakob disease (CJD) in humans[1–3]. They are devoid of nucleic acid and composed principally or entirely of multichain fibrillar assemblies of misfolded, host-encoded prion protein (PrP), a glycosylphosphatidylinositol (GPI)-anchored cell surface glycoprotein containing two asparagine (N)-linked glycosylation sites[1–3]. Prions propagate by means of seeded protein polymerization, which involves recruitment of PrP monomers to fibrillar assemblies followed by fragmentation of these elongating structures to generate more seeds. Different prion strains produce distinct clinicopathological phenotypes in the same inbred host and appear to be encoded by distinct misfolded PrP conformations and assembly states[1–3]. Since the discovery of prions, considerable international effort has been focused on determining their structure in order to understand unique facets of prion biology, including the mechanisms of replication, the differences between prions and non-infectious amyloid, the molecular basis of strain diversity, inter-species transmission barriers and toxicity[1–4]. Significantly, the generation of self-propagating polymeric or amyloid protein assemblies is now widely recognised to be involved in the pathogenesis of many other human diseases. Consequently, "prion-like" mechanisms of propagation and spread and whether the strain phenomenon is involved in phenotype have become a major research focus in the commoner neurodegenerative diseases[3,5–7] and recent advances have defined the structures of diverse self-propagating assemblies of tau[8–12], amyloid-β[13,14] and α-synuclein[15] from human brain.

The structural transition accompanying PrP monomer incorporation into infectious, protease-resistant, detergent-insoluble fibrillar prion assemblies (classically designated as PrP^Sc[1,16]) involves gross rearrangement of the protein fold[17]. While the cellular isoform of PrP (PrP^C) contains an ordered globular C-terminal domain containing three α-helices[18,19], PrP monomers within the infectious prion multimers adopt a β-strand-rich configuration[1,19] which confers protease-resistance to the C-terminal two-thirds of the protein sequence. The arrangement of β-strands, potential inclusion of other structural constituents, and the overall architecture of ex vivo prion fibrils (also referred to as prion rods[1,4,20]) have been intensely debated on the basis of various indirect, computational or low-resolution structural studies and diverse structural models have been proposed (reviewed in refs. [4,21]). Notably prion strains indicate structural heterogeneity and may constitute a cloud of diverse molecular assemblies (analogous to a viral quasispecies)[2,22] which further complicates definition of the unifying structural features of a prion. In particular, because strain-specific prion assemblies contain distinct and characteristic ratios of di-, mono- and non-glycosylated PrP[23–27], prion architectures must satisfactorily explain how such high-fidelity selection of PrP glycoforms is achieved[2,3,28].

Cryogenic electron microscopy (cryo-EM) and the recent advances in image processing (Relion)[29–31] have enabled direct, high-resolution structural studies of amyloids, fibrillar polymers defined by cross-β structure, in which misfolded protein monomers stack to form a ribbon of intermolecular β-sheets[19,32–34]. Disease-related PrP has long been known to present the tinctorial hallmarks of amyloid[20] and consistent with this, recent high-resolution cryo-EM studies suggest that protein cores of mammalian prions may generally adopt parallel in-register intermolecular β-sheet (PIRIBS) amyloid structures[35–40]. In vitro-generated fibrils from recombinant, bacterially derived, full-length human PrP[36], an N-terminal fragment thereof[37], or full-length human E196K PrP[40] form such amyloids, each

consisting of two symmetrical protofilaments. However, PrP monomers show distinct folds and distinct lateral contacts (inter-protofilament interfaces) in each of these amyloids, which indicates the structural plasticity of PrP, and thus, its potential for adopting different folds in different prion strains, but artificial in vitro polymerisation products have uncertain biological relevance. The recombinant PrP substrate is devoid of N-linked glycans and GPI-anchor which may impact the conformation of the amyloid core and its resultant biological properties. Notably, PrP amyloids formed from recombinant PrP alone are either devoid of detectable prion infectivity or have specific-infectivities far too low for meaningful structural analysis[2–4,41]. Efforts to elucidate prion structure have therefore concentrated on structural characterisation of ex vivo purified material having high specific infectivity, and recently, a cryo-EM study of the hamster 263K prion strain determined a high-resolution PIRIBS structure of PrP fibrils present in a purified, ex vivo prion sample of high specific infectivity[38,39].

In this study, we present a 2.7 Å cryo-EM structure of PrP fibrils present in preparations of infectious RML mouse-adapted scrapie prions[42,43], the most commonly studied laboratory prion strain, purified to extremely high specific infectivity[44]. In vitreous ice, we found that infectious prion rods are predominantly single-protofilament helical amyloid fibrils. The protofilaments have a unique PIRIBS conformation, with similarities to fibrils from the 263K hamster prion strain, but very different from the PIRIBS conformations of in vitro generated PrP amyloids. Notably, in distinction to purified preparations of the 263K hamster prions that were reported to consist of only single protofilaments[38,39], in our purified RML prion samples we observed that the single protofilaments coexist with twisted pairs of the same protofilaments. These findings are consistent with our previous low-resolution imaging of paired fibres in preparations of extremely high specific infectivity in which we were able to correlate structural entities with infectivity by bioassay of EM grids[28,45]. The presence of both single and paired protofilaments is intriguing and understanding their origins may be critical to elucidating the mechanism of prion propagation and selection of PrP glycoform ratios that distinguish some prion strains.

## Results

**Purified RML prion rods show both single and paired protofilament architectures.** RML prion rods from the brain of terminally-infected CD1 mice were purified using previously reported methods[44] with the exception that proteinase K (PK) rather than pronase E was used for initial digestion of brain homogenate. Purified fractions contained disease-related PrP at ~99% purity with respect to total protein, with all major SDS-PAGE silver-stained bands immuno-reactive with an anti-PrP monoclonal antibody on western blots, which showed the signature PK-resistant fragment size and PrP glycoform ratio that characterises the RML prion strain[44] (Supplementary Fig. 1). Mass spectrometry analyses of the purified rods showed that PK N-terminally truncates PrP monomers in the rods mainly at residue 88 (with minor cleavage sites at residues 80, 84 and 90) with no evidence for C-terminal truncation. PK-digested rods thereby comprise PrP monomers predominantly starting at residue 89 extending to the C-terminus with intact GPI anchor (Supplementary Fig. 1). Prion infectivity of purified samples was measured using the Scrapie Cell Assay[46] as reported previously[41,44]. The specific prion infectivity of all purified samples used in this study corresponds to ~$10^9$ mouse intra-cerebral $LD_{50}$/mg protein, consistent with our previous findings[28,44,45]. With the exception of very occasional collagen fibres, prion rods were the only visible protein structures in these samples.

The cryo-EM images of frozen-hydrated purified RML prion assemblies revealed two distinct fibrillar morphologies, ~10 nm-wide single protofilaments and ~20 nm-wide pairs of the same protofilaments (see results below). The ratio of observed single protofilaments to pairs was ~9:1 (Fig. 1a). The paired protofilaments show clear helical symmetry, with the crossover or half-pitch (180° helical turn) distance ranging from 150 to 180 nm (Fig. 1a). The single protofilaments were rarely sufficiently long to encompass the full crossover distance, but this was determined from the reconstruction to be on average slightly shorter (135 nm) than the pairs (Fig. 1a, b).

The presence of predominantly single protofilaments in our infectious RML prion rod preparations was surprising. Based on our previous negative-stain EM and AFM imaging of the purified RML samples[28,45], we interpreted the rod preparations to contain predominantly paired fibres. However, due to their helical twist the paired fibres, when viewed on a surface, alternate between wider, face-on (~20 nm) and narrower, edge-on (~10 nm) views. Guided by negative-stain EM[28,45] we considered that all narrower views (10 nm) seen in 2D projections corresponded to edge-on views of the pairs. With the insight provided by the high-resolution 3D cryo-EM, it is now apparent that the negative-stain 2D EM reflects a distribution of single and double-protofilament architectures (Supplementary Fig. 2).

**The RML protofilament has a PIRIBS structure**. We determined a 2.7 Å structure of the single RML protofilament and de novo built and refined an atomic model in the cryo-EM density based on mouse PrP sequence (residues T94-Y225) (Fig. 1b–d, Table 1 and Supplementary Figs. 3–5). Similar to fibrils from the hamster 263K prion strain[38,39], the RML protofilament displays a PIRIBS amyloid structure, with a single PrP chain contributing each rung or 'rib' to the resultant helical ribbon, with a 4.82 Å spacing between the rungs, a left-handed helical twist of −0.64°, and a crossover distance of approx. 1344 Å (Fig. 1b–f). Each visible rung comprises residues that are stabilised as part of the amyloid core, with short unresolved flexible tails at each end (PK-resistant core is slightly larger than amyloid core). The amyloid core has overall dimensions of approx. 10 × 7 nm (Fig. 1d) and can be divided into a double-hairpin N-terminal lobe and a single-hairpin C-terminal disulphide-stapled lobe (Fig. 1c and d). The extra (non-protein) densities in the N-terminal lobe are consistent with phospho-tungstate (PTA) polyanions ($[PW_{11}O_{39}]^{7-}$ at pH 7.8) used to facilitate prion purification[44,47,48] that form cage-like Keggin structures[48,49]. These bind to solvent-exposed strings of positively charged residues on the surface of the protofilaments (Fig. 1b, c, e, Supplementary Fig. 4). Extra densities in the C-terminal lobe are seen at the positions of N180- and N196-linked glycans and the flexible GPI-anchor at the C-terminus (Fig. 1c).

**Intra- and inter-molecular interactions in RML protofilaments**. Alternating polar and hydrophobic intra-chain interactions stabilise the conformation of the PrP chain in each amyloid rung (Fig. 2a) and form hydrophobic and hydrophilic columns along the fibril (Fig. 2b), similar to the way in which charges distribute along the fibril (Fig. 2c). These longitudinal interactions likely contribute to the extraordinary stability of the assembly and may play an important role in templating PrP misfolding, refolding into the prion strain-specific conformation and resistance to host clearance mechanisms. The other major inter-molecular interactions are the typical hydrogen bonds in the PIRIBS arrangement (Fig. 2d). Our model suggests that there are 15 inter-chain β-sheets in the RML fibril (Fig. 2d).

**N- and C-terminal lobes of the PrP subunits in the RML protofilament are staggered**. Considering longitudinal stacking of PrP monomers, the N- and C-terminal lobes of the chain are staggered along the amyloid fibril axis, so that each N-terminal lobe is co-planar with the C-terminal lobe of the consecutive rung (Fig. 3). Hydrophobic interactions between the N-terminal lobe's V120 and the C-terminal lobe's F174 and H176 mediate the contact between the staggered lobes (Fig. 3). Similar, but more pronounced staggering was reported for fibrils from the 263K prion hamster strain[38,39] (Fig. 4a, bottom panel), in which each N-terminal lobe contacts the C-terminal lobe of the second consecutive rung.

**Comparison of mouse RML protofilaments with hamster 263K fibrils**. When the two PrP conformations are aligned on the first two β-strands, the N-terminal lobes of the two strains look relatively similar (Fig. 4a), although the first hairpin structure in the RML conformation does not resemble a 'Greek key' – as it was named in the 263K fibril conformation[38,39] – and the N-terminal lobe of the 263K fibril appears less tightly packed than that of RML (Fig. 4b, see holes in the solvent-accessible surface), whereas the C-terminal lobes are markedly divergent (Fig. 4a). The distance between the first glycosylation site (N180/181 in the mouse/hamster sequence) is ~7 Å and it increases to ~34 Å for the second glycosylation site (N196/197 in the mouse/hamster sequence). Then the two folds become closer again between residues R207/208-Q216/217 (mouse/hamster numbering) and diverge again at their C-termini. The last residues of the ordered protein cores of RML and 263K fibrils are ~36 Å apart (Fig. 4a, top panel). The side view of the alignment (Fig. 4a, bottom panel) reveals the more pronounced inter-lobe stagger observed in the 263K fibril. This difference may be linked to the tighter helical twist of the 263K fibril (crossover distance of ~100 nm[38,39]) compared to that of the RML protofilament (crossover distance of ~135 nm).

The C-terminus and tip of the C-terminal lobe is where the two structures deviate the most. This tip appears rigid in RML protofilaments, likely due to relatively tight interactions stabilising the corresponding hairpin structure (Fig. 2a, d), whereas in the 263K fibril reconstruction, the density for the three amino acid residues at that tip (K194-E196) is missing (Fig. 4b), which indicates local flexibility or disorder. This flexibility is likely a consequence of the divergent fold in the 263K fibril C-terminal lobe compared to RML protofilaments. The C-terminal lobe of the RML protofilament continues to bind along the opposing strand in the hairpin structure, resulting in the C-terminal lobe being a single hairpin, while in the 263K fibril the chain pivots at residue Y218 to swing in the opposite direction and fold back on itself, forming a double hairpin structure (Fig. 4a–c). There, at the base of that second hairpin, residue Y226 interacts with I203 and I205, creating a gap in the first hairpin, which in turn destabilises its tip (which is the tip of the C-terminal lobe) (Fig. 4c, bottom panel). The difference in the conformation of the C-terminal lobe between the two fibrils corresponds to the differences in their host PrP sequence (Fig. 4c, top panel). These differences result in a narrower groove between the N- and C-terminal lobes in the RML PrP fibril than in the 263K hamster PrP fibril (Fig. 4b).

Notably, shortly after we posted the RML fibril structure in a preprint[50], Kraus and colleagues posted a ~3 Å structure of GPI-anchorless, underglycosylated RML fibrils (aRML) from RML prion-infected GPI-anchorless PrP transgenic mice[51]. Comparison of the cross sections of wild-type RML and aRML protofilaments show they are remarkably similar with the exception that residues 226–230 that are disordered in the wild-type RML fibril, presumably due to the presence of the GPI-anchor, are ordered in the aRML fibril. This comparison also shows that PTA has no major effect on the RML fibril fold as PTA was not used for purification of the aRML fibrils.

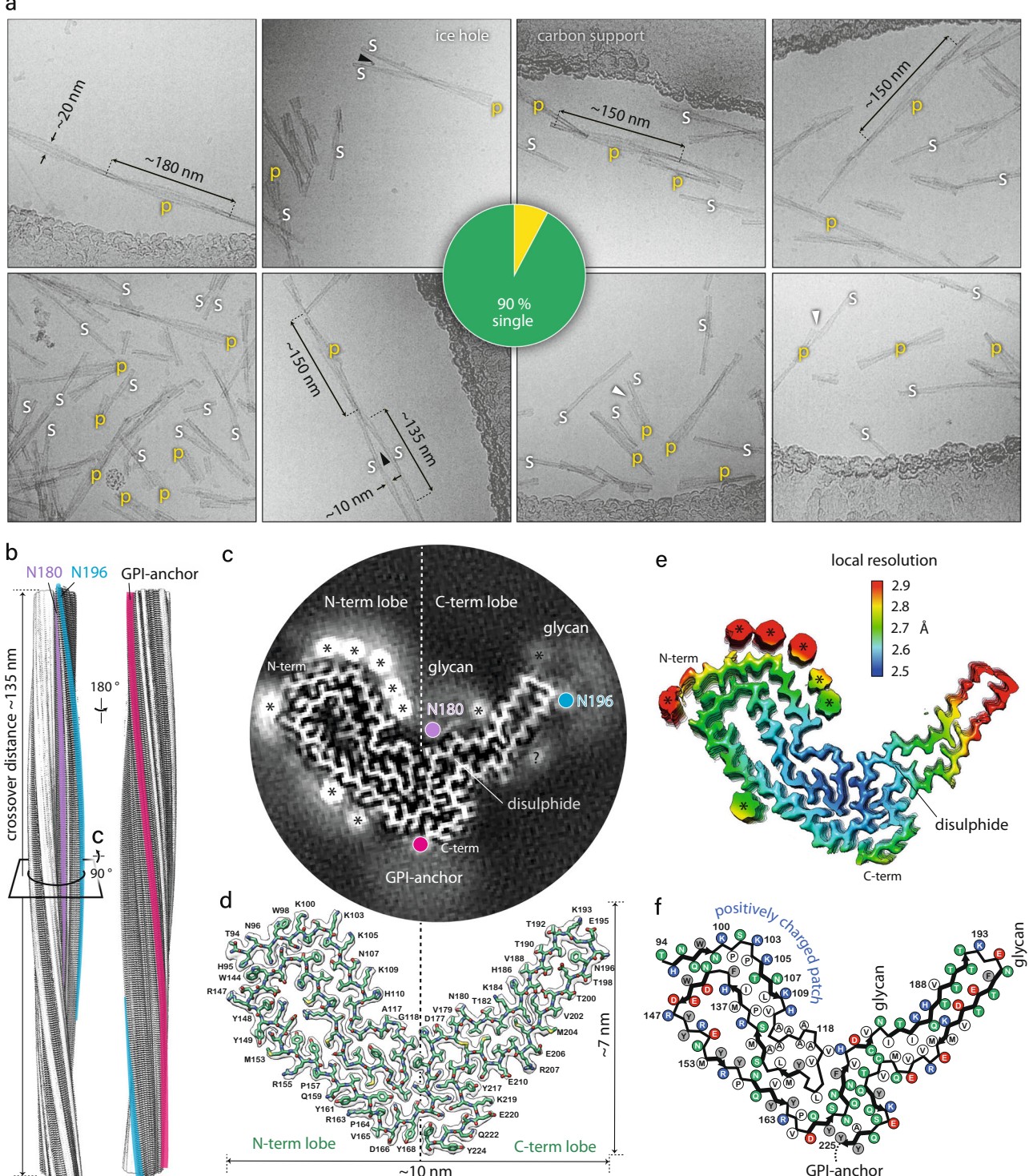

**Fig. 1 RML fibril morphologies and 3D reconstruction and atomic model of the RML protofilament core. a** Selected cryo-EM images (300 kV FEI Krios G3i, K3 camera) showing examples of single RML protofilaments (s) and paired protofilaments (p) with approximate measurements of widths and crossover distances. Black arrowheads, partly intertwined protofilaments; white arrowheads, 'notched' paired protofilaments. Pie chart, quantification of paired protofilaments in 6000 micrographs. **b** Rendered cryo-EM map (isosurface) view of both sides of a helical crossover, with annotated locations of N-linked glycans and the GPI-anchor. **c** Cross-section (as indicated in **b**) of the unrendered map (pixel size: 1.067 Å) showing the protein core and the non-protein extra densities, with annotations coloured as in **b**; *, likely locations of phosphotungstate polyanions (cages) near positively charged residues;?, unassigned density. See also Supplementary Fig. 4b. **d** Protein-only density of a single amyloid rung with the fitted atomic model of the mouse PrP chain shown with sticks and coloured by heteroatom: C, green; N, blue; O, red; S, yellow. **e** Local resolution of the map calculated with Relion 3.1 LocRes. *, as in **c**. **f** Diagram of the PrP subunit. Positions of amino acid side chains are indicated with circles (positively charged, blue; negatively charged, red; neutral, green; hydrophobic, white; aromatic, grey) on either side of the backbone (black line). β-strands are indicated with thick black arrow-headed lines.

**Table 1 Cryo-EM data collection, refinement and validation statistics.**

| | RML (EMD-13989) (PDB 7QIG) |
|---|---|
| **Data collection and processing** | |
| Magnification | 81,000x |
| Voltage (kV) | 300 |
| Electron exposure (e–/Å$^2$) | 49 |
| Defocus range (μm) | from −3.0 to −1.5 |
| Pixel size (Å) | 1.067 |
| Symmetry imposed | C1 |
| Initial particle images (no.) | 771,499 |
| Final particle images (no.) | 119,390 |
| Map resolution (Å) | 2.7 |
| FSC threshold | 0.143 |
| Map resolution range (Å) | 2.5–3 |
| **Refinement** | |
| Initial model used (PDB code) | de novo |
| Model resolution (Å) | 2.7 |
| FSC threshold | 0.143 |
| Model resolution range (Å) | 2.7–50 |
| Map sharpening B factor (Å$^2$) | −36.9 |
| **Model composition** | |
| Non-hydrogen atoms | 6255 |
| Protein residues | 396 |
| Ligands | none |
| **B factors (Å$^2$)** | |
| Protein | 31.38–66.09 |
| **r.m.s. deviations** | |
| Bond lengths (Å) | 0.005 |
| Bond angles (°) | 0.693 |
| **Validation** | |
| MolProbity score | 1.56 |
| Clashscore | 2.72 |
| Poor rotamers (%) | 0 |
| **Ramachandran plot** | |
| Favoured (%) | 91.54 |
| Allowed (%) | 8.46 |
| Disallowed (%) | 0 |

Analysis of RML paired protofilaments enabled two low resolution 3D reconstructions (Supplementary Fig. 5). These images clearly demonstrate that the paired structures contain protofilaments with the fold that we describe. However, bound PTA is close to the protofilament interface in both paired assemblies raising the possibility that PTA might be contributing to this pairing. Accordingly, we have now purified RML prion fibrils without PTA and have found that these preparations contain paired protofilaments whose morphology in ice appears very closely similar to those seen in samples prepared with PTA (Supplementary Fig. 5). These findings establish that pairing per se is not simply a PTA-induced artefact.

## Discussion

In the present study, we have determined a 2.7 Å structure of single protofilaments in highly purified preparations of infectious mouse RML prions and compared this with the recently published cryo-EM structure of fibrils in highly purified preparations of hamster 263K prions[38,39]. Both RML and 263K fibrils have a PIRIBS conformation with a PrP subunit forming each rung of the fibril, and protease-resistant cores that correspond to the sequences expected from the strain-specific signature PrP 27–30 truncated PrP$^{Sc}$ banding patterns seen on western blots. The PIRIBS conformation of prion fibrils is compatible with earlier

studies that examined ex vivo material using a variety of other techniques (comprehensively reviewed in refs. [38,39]).

The fold in the core of both RML and 263K fibrils creates distinct N- and C-terminal lobes in each PrP monomer thereby generating a broadly similar overall architecture in which the N-linked glycans and GPI anchor project from the C-terminal lobe. Significantly, the protofilament fold in RML fibrils purified from wild-type mice that we report here appears to be congruent with recently described aRML fibrils purified from RML-prion infected transgenic mice expressing GPI-anchorless PrP[51].

While the architectures of ex vivo mouse RML and hamster 263K fibrils are clearly similar, they are notably distinct from the recently reported cryo-EM structures of recombinant PrP amyloids[36,37,40] and the β-solenoid structure postulated for ex vivo PrP amyloid fibrils purified from the brain of RML-prion infected transgenic mice expressing GPI-anchorless PrP[52].

The mouse RML and hamster 263K fibrils are remarkably similar, considering the increasingly wider landscape of ex vivo amyloid structures[12–15,19,32–34], which may be relevant for the unique properties of prions. Despite this overall structural similarity, there are pronounced differences in the fold of the C-terminal lobes of mouse RML and hamster 263K fibrils, which may be attributable to differences in PrP primary structure and/or represent distinct conformational templating by divergent prion strains. To directly observe strain-specific prion conformations on the same PrP sequence, high-resolution cryo-EM structures of fibrils from other mouse- or hamster-adapted prion strains are required.

Authentic prion structures should account for the mechanism by which high-fidelity selection of PrP glycoforms is achieved during prion assembly[2,3,28]. The glycoform ratios of PrP in mouse RML fibrils and hamster 263K fibrils are very different, with marked predominance of di-glycosylated PrP in the 263K fibril[38,39] and predominance of mono-glycosylated PrPs in the RML fibril[44] (Supplementary Fig. 1). Although in silico modelling suggests no obvious steric hindrance for accommodating even solely di-glycosylated PrP chains into a cross-β amyloid structure[53], the degree of helical twist and the size of the cleft between the N- and C-terminal lobes in which residue N180/N181 (mouse/hamster numbering) is contained may favour certain PrP glycoforms over others. Notably, the 263K fibril has a much wider groove between the N- and C-terminal lobes than RML fibrils (Fig. 4b), which may sterically permit more glycan occupancy at N181. Moreover, the 263K fibril has a greater helical twist than RML fibrils (100 nm[38,39] versus 135 nm crossover distance, respectively), which may help to displace successive glycans and thereby accommodate a higher glycan occupancy at both N181 and N197 sites. These structural differences may favour incorporation of di-glycosylated PrP monomers (which are the most abundant PrP$^C$ glycoforms) into the 263K fibril over the RML fibril.

The finding that the RML single protofilament fold is very closely similar in RML-infected wild type mice and GPI-anchorless PrP transgenic mice indicates that the absence of the GPI anchor and lower levels of glycosylation do not have a major impact on the stability of this fold[51]. These data are consistent with earlier findings that GPI-anchorless RML PrP$^{Sc}$ shows very high stability in chaotropes or when heated[54]. However, it is important to note that the RML prion strain was originally isolated from wild-type mice[42,43] expressing GPI-anchored and fully glycosylated PrP and that the aRML fibril fold was templated by wild-type RML fibrils. While these new cryo-EM data show that the RML fibril fold can stably propagate in the absence of post-translational modifications, they do not inform on potentially critical roles for the GPI-anchor or N-glycans in dictating the genesis of the fold. The fact that aRML fibrils can propagate

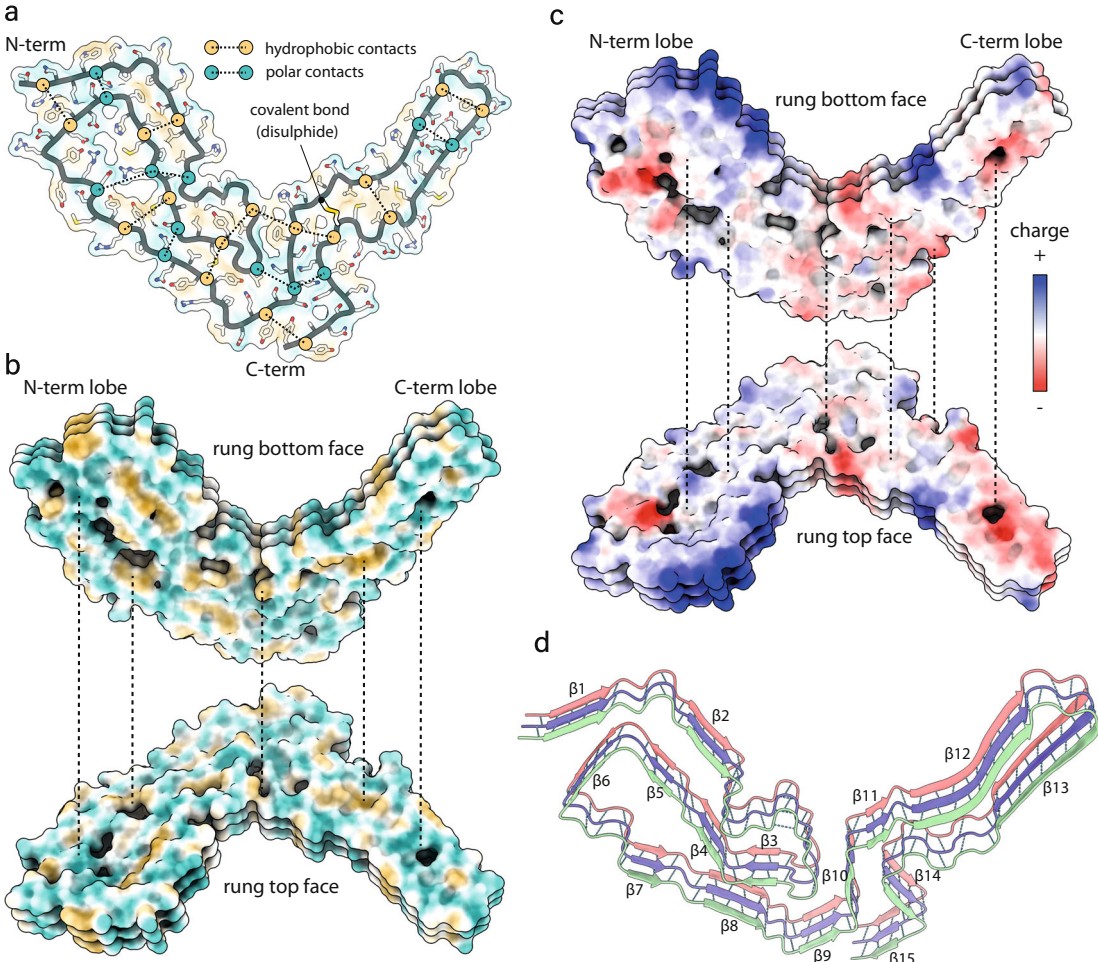

**Fig. 2 Intra- and inter-chain interactions stabilising the RML prion fibril. a** Schematic depiction of the alternating polar and non-polar lateral contacts that stabilise a single PrP monomer in the fibril. Transparent surface representation is coloured by hydrophobicity (hydrophobic, yellow; hydrophilic, teal). Protein backbone is shown with cartoon (licorice) representation and amino acid side chains as white sticks coloured by heteroatom (O, red; N, blue; S, yellow). **b** Butterfly view of the top and bottom surface of each rung, coloured as in **a**. Dotted lines indicate longitudinally connecting regions in the assembly. **c** Butterfly view of charge distribution in the assembly. Dotted lines as in **b**. **d** Ribbon representation of 3 amyloid rungs, with indicated β-sheets and inter-chain hydrogen bonds (dotted lines).

efficiently in wild-type mice[54,55] is not surprising as the RML fold at its inception would have had to sterically accommodate N-glycans and the GPI anchor. Indeed, propagation of aRML templates in wild-type mice restores the signature glycoform ratio of the RML strain[54].

While the architecture of the single RML and 263K protofilaments may be sufficient to explain generation of strain-specific PrP glycoform ratios, a potential alternative mechanism is the propagation of paired protofilaments in which the architecture of the pairs sterically limits the space available for glycan occupancy as monomers assemble into each protofilament. Within such architectures the glycans themselves may also interact with one another and contribute to the overall stability of the assembly and to the ability of prions to evade host defences. Our observation of paired protofilaments in the purified RML prion samples may suggest such a mechanism and paired protofilaments have also recently been reported in purified samples from L-type BSE-prion-infected transgenic mouse brain[56]. Importantly, while we have established that protofilament pairing per se is not simply a PTA-induced artefact, PTA may contribute heterogeneity to protofilament pairing, or conversely, PTA may disrupt the interface of paired assemblies leading to the generation of single protofilaments. How the proportions of single and double

protofilament architectures are impacted by PTA, and potential heterogeneity that PTA may contribute to pairing, are currently unknown. We are now working to obtain high-resolution cryo-EM data of PTA-free RML fibrils. From these data, it will be critical to determine the atomic structures of the paired assemblies and establish whether the single protofilaments we observe in our samples (i) originate from pairs which are the replicating species, (ii) replicate independently and then pair, or (iii) coexist with pairs as two independent seed architectures.

## Methods

**Research governance.** Frozen brains from mice with clinical prion disease were used to generate purified prion samples. These brain samples were generated by us as part of a previous study[44] in which work with animals was performed in accordance with licences approved and granted by the UK Home Office (Project Licences 70/6454 and 70/7274) and conformed to University College London institutional and ARRIVE guidelines. All experimental protocols were approved by the Local Research Ethics Committee of UCL Queen Square Institute of Neurology/ National Hospital for Neurology and Neurosurgery. Prion purification, cell-based prion bioassay and preparation of cryo-EM grids was conducted at UCL in microbiological containment level 3 or level 2 facilities with strict adherence to safety protocols. Work with infectious prion samples at Birkbeck College London was performed using dedicated sample holders and equipment with strict adherence to safety procedures and local risk assessment. Prion samples were

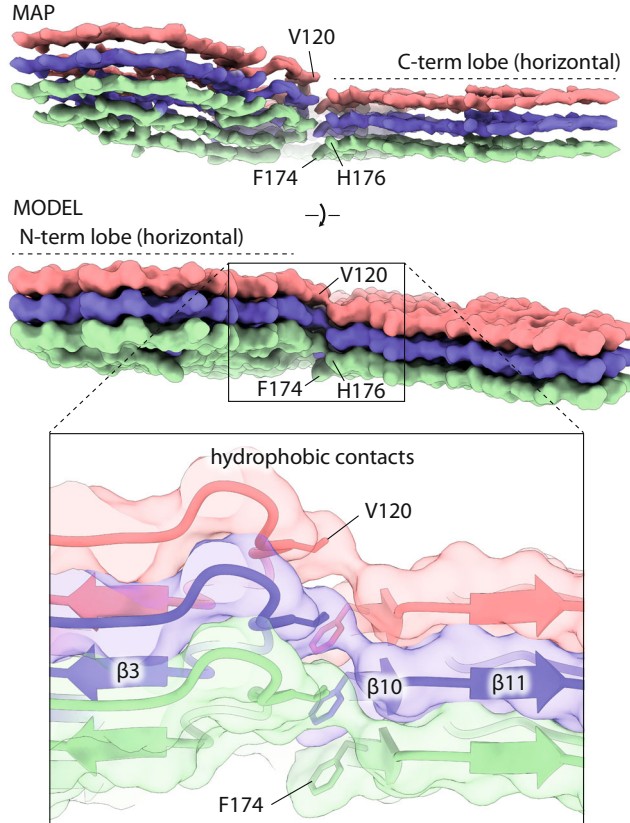

**Fig. 3 Staggering of the N- and C-terminal lobes in the RML prion fibril.** Top, cryo-EM density (MAP) and solvent-excluded surface (MODEL) of three rungs, with indicated side chains that connect to form the inter-lobe contacts. The F174 and H176 residues are better visible in the MAP and MODEL views, respectively. Bottom, magnified view of the staggered interactions (hydrophobic contacts) shown with transparent surface, main chain as ribbon and selected interacting residues as sticks.

transported between laboratories in packaging conforming to UN 3373 Biological Substance, Category B specifications.

**Preparation of purified RML prion rods**. Prion-infected brain homogenate was prepared by homogenizing 200 brains from female CD-1 mice terminally infected with the RML prion strain in Dulbecco's phosphate-buffered saline lacking $Ca^{2+}$ or $Mg^{2+}$ ions (D-PBS; Gibco) to produce a pool of ~1 l 10 % (w/v) RML brain homogenate (designated I17700) using methods described previously[44]. Purification of RML prion rods was performed as described previously[44] with the exception that initial protease digestion was performed using PK in the place of pronase E. Briefly, 200 μl aliquots of 10% (w/v) RML brain homogenate were dispensed into standard 1.5 ml microfuge tubes with screw cap and rubber O ring. Typically, 12 tubes were processed at a time. Samples were treated with 2 μl of 5 mg/ml PK prepared in water (to give 50 μg/ml final protease in the sample) and incubated for 30 min at 37 °C with constant agitation, after which digestion was terminated by addition of 4.1 μl of 100 mM 4-(2-Aminoethyl) benzenesulfonyl fluoride hydrochloride (AEBSF) to give 2 mM final concentration in the sample. 206 μl of 4% (w/v) sarkosyl (Calbiochem) in D-PBS and 0.83 μl of Benzonase (purity 1; 25,000 U/ml) were then added to give final concentrations in the sample of 2% (w/v) and 50 U/ml, respectively. Following incubation for 10 min at 37 °C, 33.5 μl of 4% (w/v) sodium phosphotungstate (NaPTA) prepared in water pH 7.4 was added to give a final concentration of 0.3% (w/v) in the sample. After incubation for 30 min at 37 °C the samples were adjusted (and thoroughly mixed) with 705.3 μl of 60% (w/v) iodixanol and 57.2 μl of 4% (w/v) NaPTA prepared in water pH 7.4, to give final concentrations in the sample of 35% (w/v) and 0.3% (w/v), respectively. After centrifugation for 90 min at 16,100 *g* the sample separates into an insoluble pellet fraction (P1), a clarified supernatant (SN1) and a buoyant, partially flocculated, surface layer (SL). 1 ml of SN1 was carefully isolated from each tube taking extreme care to avoid cross contamination with either P1 or SL. SN1 was filtered using an Ultrafree-HV microcentrifuge filtration unit (0.45 μm pore size Durapore membrane, Millipore, Prod. No. UFC30HV00). This was accomplished by loading 500 μl aliquots of SN1 and centrifugation at 12,000 *g* for 30 sec using one filtration unit per ml of SN1. 480 μl aliquots of filtered SN1 were transferred to new 1.5 ml microfuge tubes and thoroughly mixed with an equal

volume of 2% (w/v) sarkosyl in D-PBS containing 0.3% (w/v) NaPTA pH 7.4 and incubated for 10 min at 37 °C. Samples were then centrifuged for 90 min at 16,100 *g* to generate an insoluble pellet fraction (P2) and a clarified supernatant (SN2). SN2 was carefully removed and discarded, after which each P2 pellet was resuspended in 10 μl of 5 mM sodium phosphate buffer pH 7.4 containing 0.3% (w/v) NaPTA and 0.1% (w/v) sarkosyl. In order to avoid unnecessary aggregation of the purified rods arising from repeated rounds of centrifugation the final two wash steps detailed in Wenborn et al. 2015[44] were replaced with a single wash. Resuspended P2 pellets were pooled and mixed with 1.0 ml of 5 mM sodium phosphate buffer pH 7.4 containing 0.3% (w/v) NaPTA and 0.1% (w/v) sarkosyl and samples centrifuged at 16,100 *g* for 30 min to generate a clarified supernatant (SN3) and an insoluble pellet fraction (P3). SN3 was carefully removed and discarded and final P3 samples were typically resuspended to a concentration of 120X relative to the starting volume of 10 % (w/v) brain homogenate from which they were derived, prior to loading on to EM grids (see below). Purification of RML prion rods without NaPTA was performed as described above with the following modifications. Additions of 4% (w/v) NaPTA pH 7.4 were replaced with equivalent volumes of water. Filtered SN1 was diluted 3-fold in 2% (w/v) sarkosyl in D-PBS to give 11.67% (w/v) iodixanol in the sample prior to centrifugation to generate P2 pellets. Final resuspension of the P3 pellets was done in 50 mM tris containing 150 mM NaCl pH 7.4, prior to loading on to EM grids.

Prion infectivity of brain homogenates or purified samples was measured using the Scrapie Cell Assay (SCA) and Scrapie Cell End Point Assay (SCEPA)[46] using PK1/2 cells (an established cell line; D-Gen Ltd, London). Every SCA experiment included concomitant assay of a serial dilution of RML prions of known prion titre determined from rodent bioassay to produce a standard curve that unknown samples were calibrated against. 10 % (w/v) RML brain homogenate I6200 was used as the standard and reported a prion titre of $10^{7.3 + 0.5}$ (mean + s.d.) intracerebral $LD_{50}$ units/ml when endpoint titrated six times in Tg20 mice that overexpress mouse PrP on a $Prnp^{o/o}$ background, corresponding to $10^{7.7}$ TCIU/ml in PK1/2 cells[44]. PrP concentrations in purified samples were measured by ELISA as described previously[44].

**SDS-PAGE, silver staining and western blotting**. Samples were prepared for SDS-PAGE using NuPage 4X LDS buffer and 10X Reducing Agent (Thermo Fisher) according to the manufacturer's instructions followed by immediate transfer to a 100 °C heating block for 10 min. Electrophoresis was performed on NuPage 12 % Bis-Tris protein gels (Thermo Fisher), run for 60 min at 200 V, prior to electroblotting to Immobilon P membrane (Millipore) for 16 h at 15 V. Membranes were blocked in 1X PBS (prepared from 10X concentrate; VWR International) containing 0.05% (v/v) Tween 20 (PBST) and 5% (w/v) non-fat dried skimmed milk powder and then probed with 0.2 μg/ml anti-PrP monoclonal antibody ICSM35 (D-Gen Ltd) in PBST for at least 1 h. After washing (1 h with PBST) the membranes were probed with a 1:10,000 dilution of alkaline-phosphatase-conjugated goat anti-mouse IgG secondary antibody (Sigma-Aldrich, Cat No A2179) in PBST. After washing (1 h with PBST and 2 × 5 min with 20 mM Tris pH 9.8 containing 1 mM $MgCl_2$) blots were incubated for 5 min in chemiluminescent substrate (CDP-Star; Tropix Inc) and visualized on Biomax MR film (Carestream). SDS-PAGE gels (prepared as above) were silver stained using the Pierce Silver Stain Kit (Thermo Fisher) according to the manufacturer's instructions. Gels were photographed on a light box using a Nikon Coolpix P6000 digital camera. Typical sample loadings for western blotting or silver staining correspond to purified material derived from 10 μl or 100 μl of 10 % (w/v) RML prion-infected brain homogenate per lane, respectively. The SDS-PAGE and western blot data generated in this study are provided in a Source Data file.

**Determination of N-terminal PK-cleavage sites in purified RML prion rods by mass spectrometry**. N-terminal PK-cleavage sites in PrP were determined by the targeted derivatization of α-amino groups and subsequent analysis by mass spectrometry, as described by Deng et al. (2015)[57] with minor modifications. Briefly, purified RML fibils were prepared as above and subjected to SDS-PAGE separation using NuPAGE 12% Bis-Tris mini protein gels (Thermo Fisher) according to the manufacturer's instructions. Gel sections spanning all three PrP glycoforms were excised, reduced with 100 μM tris(2-carboxyethyl)phosphine and alkylated with 200 μm iodoacetamide prior to N-terminal labelling with 6 mM N-Succinimidyloxycarbonylmethyl tris(2,4,6-trimethoxyphenyl)phosphonium bromide, (TMPP-Ac-OSu) (Sigma) for 1 h at 22 °C in 100 mM HEPES buffer pH 8.2. Gel-pieces were then washed thoroughly and subjected to overnight trypsin digestion at a working concentration of 2.5 μg/ml. The following day, tryptic digest peptides were recovered by gel extraction, as described by Shevchenko et al. (2006)[58]. Peptide analysis was performed by liquid chromatography mass-spectrometry, using an Acquity I-Class UPLC system coupled to a Xevo G2-XS Q-ToF mass spectrometer (Waters). Data was collected in MSe acquisition mode using concurrent low- and high-collision energy functions with 5 V and 15–45 V of collision energy, respectively. Peptide sequence assignments were made using ProteinLynx Global Server 3.0.3 (Waters) against a species-specific reference proteome (Uniprot UP000000589, mus musculus) and optionally allowing for N-terminal amino-group derivatization by TMPP (+572.1811 Da). Extracted ion chromatograms were generated for each TMPP-labelled peptide and relative abundance was determined from their respective peak areas. These data are provided in a Source Data file.

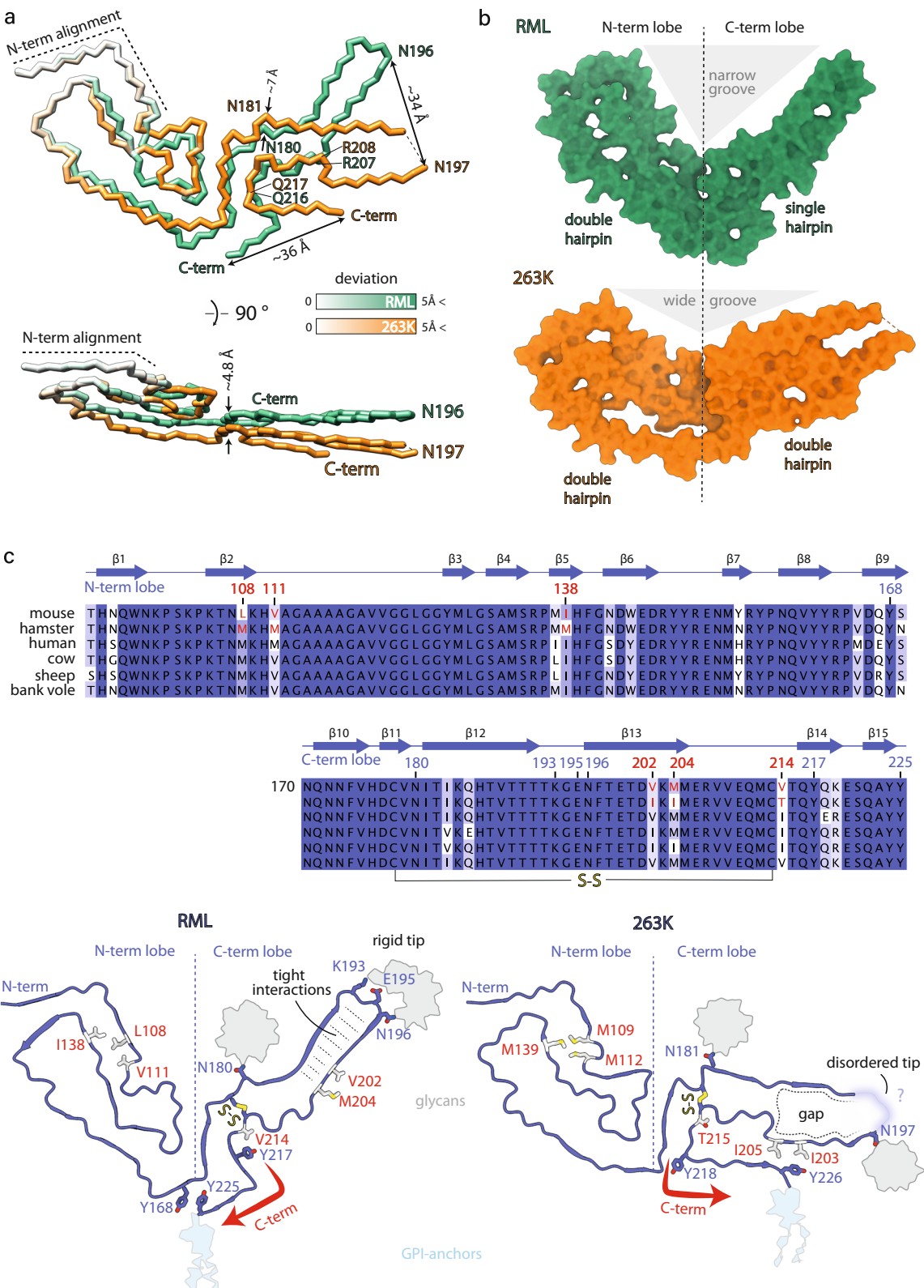

**RML sample preparation for cryo-EM**. RML prion rods purified from 2.4 ml 10 % (w/v) RML-infected brain homogenate were resuspended from the P3 pellet (see above) in 20–30 μl 5 mM sodium phosphate buffer pH 7.4 containing 0.1 % (w/v) sarkosyl and 4 μl of the suspension was applied directly to a glow-discharged C-flat™ Holey Carbon CF-2/2–4 C Cu 400 mesh cryo-EM grid (Electron Microscopy Sciences) in the chamber of the Leica GP2 plunging robot. The chamber was set to 20 °C and 50% humidity. After 10 s incubation, the grids were blotted for 3 s (with an additional 2 mm push) and plunge-frozen in liquid ethane maintained at −183 °C.

**Cryo-EM data collection**. Cryo-micrographs were acquired at Birkbeck College London, on a 300 kV Krios G3i microscope (FEI/Thermo Fisher) with a post-GIF (20 eV slit) K3 detector (Gatan) operated in super-resolution mode. The magnified pixel size was 0.5335 Å. The dose rate was 16.37 e-/Å²/s during 3-s exposures, resulting in the total dose of 49 e-/Å² on the specimen. The exposures were collected automatically at four shots per grid hole, with fast acquisition (240 images/hr), using the EPU 2 software (FEI/Thermo Fisher), at defocus ranging from −3.0 to −1.5, and fractionated into 50 movie frames.

**Fig. 4 Comparison of PrP conformation in the RML and the 263K prion fibrils. a** Polypeptide backbone superposition on the first two β-strands (N-term alignment; secondary structure not shown) of single PrP monomers from the two different strains, coloured by their deviation in distance. **b** Surface models showing internal gaps and the divergent angles between the N- and C-terminal lobes. **c** Top, multiple PrP sequence alignment coloured by conservation and annotated by mouse RML PrP sequence numbering and secondary structure. Mouse vs hamster amino acid substitutions (AAS) that underpin distinct conformations of RML and 263K fibrils are highlighted in red. Bottom, mapping of the selected AAS onto cartoon structures of RML (this study) and 263K (PDB code: 7LNA) fibrils (sticks coloured white and by heteroatom: O, red; S, yellow). Selected conserved residues, including those involved in distinct interactions due to divergent PrP folds are shown with sticks coloured as main chain and by heteroatom (N, blue; O, red; S, yellow). Red arrows indicate different folds of the C-termini, which result in divergent tips of the C-terminal lobes.

**Cryo-EM image processing and 3D reconstruction**. All image processing except particle picking was done within the framework of Relion 3.1[31]. We used Relion's implementation of the MotionCor2 algorithm to align movie frames. The images were 2x binned in Fourier space during the frame alignment, resulting in the final pixel size of 1.067 Å$^2$ in the drift-corrected sums. The contrast transfer function (CTF) parameters were estimated with Gctf[59]. We then picked particles (fibril segments) using the deep learning package crYOLO[60,61] trained on 100 example micrographs. The picking was accurate and avoided crowded regions with overlapping or clumped rods, fibrils on carbon support and fibrillar bundles, as illustrated in Supplementary Fig. 3. We imported the coordinates into Relion and extracted images of prion rod segments of different box sizes (ranging from 1024 to 384 pixels) to perform reference-free 2D classification. Optimal 2D class averages and segments were selected for further processing and used to de novo generate an initial 3D reference with relion_helix_inimodel2d programme[31], using an estimated rise of 4.75 Å and helical twist according to the observed crossover distances of the filaments in the 2D class averages. After 3D classification and 3D autorefinement, we obtained a 3D reconstruction of the RML fibril at 3.0 Å resolution in a 384-pixel cube. Subsequent Bayesian polishing[62] and CTF refinement[63] were performed to further improve the resolution of the reconstruction to 2.7 Å, according to 0.143 FSC cut-off criterion (Supplementary Fig. 6). The final 3D map was sharpened with a generous, soft-edged solvent mask at 10% of the height of the box using the computed B-factor value of −36.9 Å$^2$. The sharpened map was used for the subsequent atomic model building and refinement. The absolute hand of the helical twist was determined directly from the map through resolved densities of the carbonyl oxygen atoms of the polypeptide backbone[31]. The local resolution calculation was performed by LocRes in Relion 3.1 with solvent mask over the entire map. Paired fibrils were picked manually and processed as described above. The data supported low resolution 3D reconstruction of two types of paired assemblies, but 2D classifications suggest that other modes of pairing may also be present.

**Atomic model building and refinement**. A single subunit repeat was extracted in UCSF Chimera[64] for the initial de novo model building in Coot[65]. The initial atomic model was then copied and fitted into their consecutive subunits in the map and the map was zoned around the atomic coordinates in UCSF Chimera[64]. The 3-rung map and model were placed in a new unit cell with P1 space group for subsequent model refinement using default settings in phenix.real_space_refine[66] and REFMAC5[67] with non-crystallographic symmetry (NCS) group definitions constraining the helical subunit repeat. Model geometry was evaluated using the MolProbity server[68] (http://molprobity.biochem.duke.edu/) after each round of refinement, and problematic or poorly fitting regions in the model were manually adjusted using Coot[65] and Isolde[69] (within ChimeraX[70]). This process was repeated until a satisfactory level of model:map agreement with acceptable model stereochemistry was achieved (Table 1).

**Negative-stain EM**. RML prion rods purified from 2.4 ml 10 % (w/v) RML-infected brain homogenate were resuspended from the P3 pellet (see above) in 40 µl 50 mM tris, 150 mM NaCl pH 7.4 (TBS) containing 0.1% (w/v) sarkosyl and deposited on glow-discharged EM grids with a continuous carbon film (Agar). The grids were briefly blotted and washed with TBS before staining with 2% solution of NANO-W$^{TM}$ stain (Nanoprobes). After ~1 s exposure to the stain solution, the grids were blotted again and air-dried. The negatively stained grids were imaged in the Unit on a 120 kV Talos microscope (FEI/Thermo Fisher) with a 4k × 4k BM-Ceta camera.

**Structure analyses and presentation**. Analyses and visualisations of the cryo-EM density map and the models compared in this study were done using UCSF Chimera[64] and ChimeraX[70].

**Statistics and reproducibility**. Purification of RML prions was successfully replicated ~20 times while optimizing sample concentrations for cryo-freezing using the Talos microscope. Eleven cryo-EM grids containing material from five independent prion purifications were used for data collection in the Krios G3i microscope. Representative images of prion rods in ice were selected from a data-set comprising ~6000 multi-frame movies.

**Reporting summary**. Further information on research design is available in the Nature Research Reporting Summary linked to this article.

## Data availability

The cryo-EM data generated in this study have been deposited in the EMPIAR database under accession code EMPIAR-10992. The 3D cryo-EM density map was deposited into the Electron Microscopy Data Bank (https://www.ebi.ac.uk/pdbe/emdb) under accession code EMD-13989 (Infectious mouse-adapted RML scrapie prion fibril purified from terminally-infected mouse brains). The corresponding atomic coordinates were deposited in the Protein Data Bank (https://www.rcsb.org) under PDB code 7QIG. Uncropped and unprocessed SDS-PAGE and western blot data and mass spectrometry data generated in this study are provided in Source Data files.

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

## Acknowledgements

This work was funded by the core award to the MRC Prion Unit from the UK Medical Research Council (MC_U12316055 and MC_UU_00024/5 to J.D.F.W.). EM data collection was supported by grants from the Wellcome Trust (202679/Z/16/Z, 206166/Z/17/Z, 106249/Z/14/Z to H.R.S.). We are very grateful to Dr Natasha Lukoyanova and Dr Shu Chen at Birkbeck College for EM support and Damian Johnson, Peter King, Kevin Williams and Kevin Foulger for infrastructure support at UCL. We would like to dedicate this paper to the late Professor Anthony Clarke who founded the protein structure programme at the MRC Prion Unit and who made many fundamental contributions to this field.

## Author contributions

J.C., J.D.F.W. and H.R.S. conceived the project, acquired funding and oversaw the study. J.D.F.W. administered the project. S.W.M. and J.D.F.W. supervised the research. S.W.M., A.W., J.B. and J.D.F.W. designed experiments. S.W.M., A.W. and J.B. performed experiments. S.W.M., W.Z. and J.B. processed data. S.W.M., W.Z., A.W. and S.J. analysed data. W.Z. reconstructed and refined the cryo-EM map. S.W.M. built and refined the atomic model. S.W.M., A.W., W.Z., H.R.S., J.D.F.W. and J.C. interpreted results. S.W.M. and J.D.F.W. wrote the manuscript with contributions from all authors.

## Competing interests

J.C. is a Director and J.C. and J.D.F.W. are shareholders of D-Gen Limited, an academic spin-out company working in the field of prion disease diagnosis, decontamination, and therapeutics. D-Gen supplied the ICSM35 and ICSM18 antibodies used for western blot and ELISA performed in this study. The other authors declare no competing interests.
