## [Peer Review File · Nature Communications]

REVIEWER COMMENTS

Reviewer #1 (Remarks to the Author):

Manka et al. present a remarkably detailed view of the RML prion fibril. In this tour de force, they build a model for the fibril de novo, with hints at the locations of glycans, a clear map of its core. Their structure points also illuminates differences vs. 263K fibrils, particularly in its c-terminal portion. This is the second in an important collection of structures that will help reveal the atomic basis for prion infectivity.

The authors claim to have prepared the RML rods (fibrils) with exacting precision to high titre using a PK digestion protocol, but oddly also note that the details of their analysis of this process will be presented elsewhere. Why would the details not be included, at least as a supplement, to the study describing this structure? They seem critical here.

The authors claim to observe two distinct fibrillar morphologies, single and paired protofilaments, which they imply have the same underlying structure. Yet, the two fibril species have differing half-pitch values and possibly different structures (even if largely similar). A structure of the paired protofilaments at any reasonable resolution would help resolve this ambiguity.

In addition, the presence of phosphotungstate clusters bound to the outer face of the reconstructed fibril is notable. Several sites of cluster binding are noted, coating a significant portion of the fibril surface. Are these molecules not expected to impact the observed proportions of single and double protofilaments measured by the authors?

The authors note that their fibrils showed a left-handed helical twist of -0.64° and a crossover distance of $\sim 1344 \text{ \AA}$. How was the handedness of the helical twist determined?

The authors note that the fibrils used in their reconstruction were shorter than their anticipated full crossover distance. Could the authors discuss or possibly speculate as to why this happened to be the case for RML prions? Is this expected to be a function of the purification method, or might there be something inherent to this structure that limits fibril length?

Could the authors further comment on the specific relevance of this structure to particular mechanisms of disease propagation beyond glycoform selection? What about the features of the structure that help endow the core with its unique stability? Are the glycans serving a role in the packing of the core - eg. might better ordering of the c-terminal region here be influenced by the presence of particular glycans?

The authors claim that structures of recombinant, bacterially derived prion fibrils have uncertain if any biological relevance, in part because they lack glycans and a GPI anchor. However, the authors later point to the similarity of their structure to one derived from mice expressing GPI-anchorless PrP. Is the GPI anchor an absolute requirement for relevance?

This illustrates the importance of a more nuanced discussion over relevance, rather than such a black and white dichotomy in favor of tissue-derived infectious prions focusing on a couple of chemical moieties.

While structural comparisons are valuable, wouldn't the authors agree that focusing so sharply on what they believe are perceived 'pronounced' differences between their structure and the previously published hamster 263K structure or GPI anchorless RML structure, might limit a broader discussion?

Sure the structures are different in part, but to this reviewer, it is breathtaking to see that two prion strains with different characteristics derived from different species contain roughly the same fibril structures (at least half of which is extremely similar). There may be something truly remarkable about this fold that could teach us something about why prions have such unique properties within the realm of amyloids.

Lastly, the authors might further improve their model geometry while maintaining a good fit to map by relaxing the model in PyRosetta with density and symmetry restraints.

Reviewer #2 (Remarks to the Author):

2.7A cryo-EM structure of ex vivo RML prions

Manka et al. analyzed the cryo-EM structure of RML prion rods. This is the second report of an $\sim 3\text{\AA}$ resolution structure of prion rods isolated from the brains of experimentally infected rodents with laboratory adapted scrapie prions. In light of the unique mechanism of prion replication that occurs in the absence of informational nucleic acids, solving the structure of the infectious conformation of PrP is a highly significant pursuit. Unlike the normal cellular form of PrP, the solubility of which makes it amenable to structural characterization by conventional techniques, the aggregating properties and insolubility of PrP^{Sc} have been significant impediments. Recent cryo-EM and associated computational advances provide opportunities to assess the structural properties of amyloid fibrils present in the brains of prion infected animals. This paper provides this information for amyloid resulting from infection with a second prion preparation. The findings of this paper support a hypothesis that mammalian prions may share a general PIRIBS architecture, with some regional variations reflecting the prion strain phenomenon. Whereas the first reported structure was of 263K/Sc237 prions isolated from Syrian hamster brains, the current paper reports the structure of amyloid preparations from RML prions isolated from the brains of infected mice.

One general conceptual drawback to all such studies is whether or not the analyzed structures really represent the structure of infectious prions. In time answers will come from independent structural approaches which either support or contradict the cryo-EM models of amyloid. With this caveat in mind, the authors have done a good job of correlating/tracking infectivity of purified prion preparations used for analyses using the scrapie cell assay. Also important in this regard is the fact that the overall structures of RML amyloid and that of 263K prions are similar. This finding is therefore an important aspect of the current paper. If for no other reason, this paper and its 263K predecessor are important for being able to discount the relevance of previously published putative prion structures such as the beta solenoid.

Notwithstanding these strengths, the authors report the presence of single and paired amyloid fibrils in RML preparations. This finding is reproducible since it is in keeping with previous analyses of RML from this lab, but apparently different from the situation with 263K hamster prions. At first blush this observation might lend support to the notion that the unpaired filaments represent the infectious prion structure, except for the fact that previous publications from this lab argued for the importance of the paired filaments. In fairness, the authors acknowledge this discrepancy and the importance of resolving it.

The paper is very well written and the data are clearly presented.

Additional points:

- **Mass spec indicated that PK N-terminally truncates the rods at 88 with no C-terminal truncation. The cryo-EM data concerns residues 94- 225 – please clarify this apparent discrepancy.**
- **Could the authors compare their findings suggesting a PIRIBS structure with previous independent analytical approaches to prion tertiary structures such as NMR, mass spectrometry, and hydrogen-deuterium exchange? Do those findings support the PIRIBS structure?**

2.7 Å cryo-EM structure of ex vivo RML prion fibrils
Nature Communications manuscript NCOMMS-22-00142-T

Point-by-point response to Reviewers.

All changes in the text of the manuscript have been highlighted in red ink.

Reviewer 1

We are very grateful to the Reviewer for their insightful suggestions and kind remarks stating, "*Manka et al. present a remarkably detailed view of the RML prion fibril. In this tour de force, they build a model for the fibril de novo, with hints at the locations of glycans, a clear map of its core.*" "*This is the second in an important collection of structures that will help reveal the atomic basis for prion infectivity.*"

The Reviewer had several points to address and we have revised the text accordingly. We feel that the manuscript has been significantly improved with these changes.

1. The authors claim to have prepared the RML rods (fibrils) with exacting precision to high tire using a PK digestion protocol, but oddly also note that the details of their analysis of this process will be presented elsewhere. Why would the details not be included, at least as a supplement, to the study describing this structure? They seem critical here.

We thank the Reviewer for requesting this clarification. The text that the Reviewer mentions actually refers to our mass spectrometry analyses of prion rods rather than methods used for prion purification. The text read "Mass spectrometry analyses of the purified rods showed that PK N-terminally truncates PrP monomers in the rods at residue 88 with no evidence for C-terminal truncation. PK-digested rods thereby comprise PrP monomers starting at residue 89 extending to the C-terminus with intact GPI anchor. Details of these analyses will be published elsewhere."

However we do agree with the Reviewer that the methods could be expanded and so have revised the text in the methods describing prion purification so that there is less reliance on referring to our previously published work. In addition we have also added a new section in the methods providing details of our mass spectrometry analyses to address point 1 raised by Reviewer 2. We hope that these changes are acceptable.

2. The authors claim to observe two distinct fibrillar morphologies, single and paired protofilaments, which they imply have the same underlying structure. Yet, the two fibril species have differing half-pitch values and possibly different structures (even if largely similar). A structure of the paired protofilaments at any reasonable resolution would help resolve this ambiguity. In addition, the presence of phosphotungstate clusters bound to the outer face of the reconstructed fibril is notable. Several sites of cluster binding are noted, coating a significant portion of the fibril surface. Are these molecules not expected to impact the observed proportions of single and double protofilaments measured by the authors?

We thank the Reviewer for these insightful questions regarding the structure of the paired protofilaments and whether phosphotungstate (PTA) might influence the proportions of single and double protofilaments. These are indeed key questions.

In vitreous ice we observed multiple examples of single protofilaments intertwining to form paired assemblies. We show examples of those events in Fig. 1a (black and white arrowheads). The twist angle is somewhat flexible even in single protofilaments (the value given in the reconstruction is the average value) and in the paired protofilaments the twist might be expected to be even more flexible as the pairing interface might act as a hinge. However we currently do not know the pairing mechanism. As a polyoxometalate, PTA might be capable of acting as a bridge to mediate protofilament pairing, or conversely, PTA may disrupt the paired protofilament interface leading to the generation of single protofilaments.

Analysis of RML paired protofilaments enabled two low resolution 3D reconstructions, shown in Supplementary Figure 5. These images clearly demonstrate that the paired structures contain protofilaments with the fold that we describe. However, bound PTA is close to the protofilament interfaces in both assemblies raising the possibility that PTA might be contributing to this pairing. Accordingly, we have now purified RML prion fibrils without PTA (new text added in Methods) and have found that these preparations contain paired protofilaments (shown in Supplementary Figure 5) whose morphology in ice appears very closely similar to those present in samples prepared with PTA. These findings establish that pairing *per se* is not simply a PTA-induced artefact.

To understand exactly how the proportions of single and double protofilament architectures are impacted by PTA, and the heterogeneity that PTA may contribute to pairing, we are now working to obtain further high resolution cryo-EM data sets of RML prions purified with and without PTA. This new research is ongoing and is very time consuming as we require very high particle numbers to explore paired fibril heterogeneity. In this regard, at present we have no additional data that could be added to this manuscript. Instead we have added new text to the Results and Discussion that summarise the various points discussed above. We hope that these changes are acceptable.

3. The authors note that their fibrils showed a left-handed helical twist of -0.64° and a crossover distance of $\sim 1344 \text{ \AA}$?. How was the handedness of the helical twist determined?

We thank the Reviewer for raising this important point. It is true that cryo-EM reconstruction does not always provide information on the absolute hand, however, at resolutions better than 2.9 \AA the handedness may be inferred directly from the map, since the main chain oxygen atoms become visible. We have added new text in the Methods section to clarify this point "The sharpened map was used for the subsequent atomic model building and refinement. The absolute hand of the helical twist was determined directly from the map through resolved densities of the carbonyl oxygen atoms of the polypeptide backbone (ref 41). The local resolution calculation was performed by LocRes in Relion 3.1 with solvent mask over the entire map." We hope this change is acceptable.

4. The authors note that the fibrils used in their reconstruction were shorter than their anticipated full crossover distance. Could the authors discuss or possibly speculate as to why this happened to be the case for RML prions? Is this expected to be a function of the purification method, or might there be something inherent to this structure that limits fibril length?

We thank the Reviewer for raising this point. Indeed, not all fibrils in our micrographs encompass the full crossover distance. It is possible that shorter fibril fragments could result from either the purification method or something inherent, but at present we feel we do not have enough information to unequivocally comment on this. However in this regard, we would like to point out that filtration (0.45 μm pore size) is used during prion purification which may limit fibril length in the purified sample. Changes to the text in the methods section (made in response to point 1 above) now provide details of this filtration step.

5. Could the authors further comment on the specific relevance of this structure to particular mechanisms of disease propagation beyond glycoform selection? What about the features of the structure that help endow the core with its unique stability? Are the glycans serving a role in the packing of the core - eg. might better ordering of the c-terminal region here be influenced by the presence of particular glycans?

We thank the Reviewer for these fascinating questions however we lack data to comment further on various possible mechanistic consequences of the structure. In the text we write that in addition to the cross-beta hydrogen bonds, the alternating hydrophobic contacts that propagate along the fibril likely have an important role in maintaining the extraordinary stability of the rods. We also discuss the potential role of sugars in prion rod stability, but we cannot comment more, as we are unable to resolve the densities of glycans (due to their heterogeneity and/or flexibility) and thus we are unable to pinpoint their identity in this study. It has been shown computationally that even the maximum occupancy of sugars can probably be accommodated in the PIRIBS architecture (ref 61, Artikis et al, 2020, ACS Chem Neurosci), but whether the sugars may have an additional stabilising role would depend on their exact arrangement, which at present remains elusive. Alternatively, it is also possible that N-glycan occupancy simply precludes certain misfolding pathways that recombinant PrP can follow, either by preventing formation of particular core amyloid folds or by blocking inter-prot filament interfaces.

Given all these possibilities and no additional data we have not changed the text in response to this point. We hope that the Reviewer will accept our reluctance to speculate.

6. The authors claim that structures of recombinant, bacterially derived prion fibrils have uncertain if any biological relevance, in part because they lack glycans and a GPI anchor. However, the authors later point to the similarity of their structure to one derived from mice expressing GPI-anchorless PrP. Is the GPI anchor an absolute requirement for relevance? This illustrates the importance of a more nuanced discussion over relevance, rather than such a black and white dichotomy in favor of tissue-derived infectious prions focusing on a couple of chemical moieties.

We thank the Reviewer for raising these points. We agree that our text should be less categorical and have changed the text in the Introduction and Results accordingly.

With regard to the GPI anchor and the N-glycans, as the Reviewer will be aware, Byron Caughey and colleagues posted a preprint reporting a ~ 3 Å structure of GPI-anchorless, under-glycosylated RML fibrils (aRML) (Hoyt et al 2021, BioRxiv) eight days after we posted a preprint of the current manuscript (Manka et al 2021 BioRxiv). The protofilament fold of aRML fibrils appears to be very closely similar to the fold of the RML protofilament from wild-type mice. In relation to point 2 above, these new data also show that the PTA is not perturbing the fold because PTA was not used in purification of the aRML fibrils.

The finding that the RML single protofilament fold is very closely similar in RML-infected wild type mice and GPI-anchorless PrP transgenic mice indicates that the absence of the GPI anchor and lower levels of glycosylation do not have a major impact on the stability of this fold (Hoyt et al BioRxiv 2021). These data are consistent with earlier findings that GPI-anchorless RML PrP^{Sc} shows very high stability in chaotropes or when heated (Bett et al Plos Pathog. 2013). However it is important to note that the RML prion strain was originally isolated from wild-type mice expressing GPI-anchored and fully glycosylated PrP and that aRML was templated by wild-type RML prions. While these new cryo-EM data show that the RML fibril fold can stably propagate in the absence of post-translational modifications, they do not inform on potentially critical roles for the GPI-anchor or N-glycans in dictating the genesis of the fold. The fact that aRML fibrils can propagate efficiently in wild-type mice (Chesebro et al, Science 2005, Plos Pathog. 2010; Bett et al Plos Pathog. 2013) is not surprising as the RML fold at its inception would have had to sterically accommodate N-glycans and the GPI anchor. Indeed, propagation of aRML templates in wild-type mice restores the signature glycoform ratio of the RML strain (Bett et al Plos Pathog. 2013).

Removal of the GPI anchor is not without effect however, as RML prion-infected GPI-anchorless PrP mice not only propagate authentic prions but also develop intense PrP amyloid deposits throughout their brain which are not seen in RML prion-inoculated wild-type mice (Chesebro, et al. Science 308, 1435–1439, 2005 and Plos Pathog. 6, e1000800, 2010). Following a low-resolution cryo-EM study of amyloid fibrils isolated from these mice, (Vazquez-Fernandez, et al. (PLOS Pathog; 2016) postulated a β -solenoid model. We subsequently suggested that these findings might be attributable to co-propagation of an additional, structurally distinct PrP amyloid in these mice (Terry et al Sci Rep 2019). Our data and those from Caughey and colleagues now establish that aRML fibrils have a structure congruent with RML fibrils from wild-type mice. The basis for the PrP fibril architecture proposed by Vazquez-Fernandez et al. has yet to be resolved. Caughey and colleagues are of the view that Vazquez-Fernandez and colleagues isolated aRML fibrils but then misinterpreted their low resolution cryo-EM data (Kraus et al 2021 Mol Cell). However this explanation is not entirely satisfactory as the fibrils studied by Vazquez-Fernandez and colleagues had a very low specific prion infectivity.

In light of these new structural data on aRML fibrils (that were unavailable when drafting the original manuscript) we have now added appropriate text in the Results and Discussion. We hope that these changes are acceptable.

7. While structural comparisons are valuable, wouldn't the authors agree that focusing so sharply on what they believe are perceived 'pronounced' differences between their structure and the previously published hamster 263K structure or GPI anchoress RML structure, might limit a broader discussion? Sure the structures are different in part, but to this reviewer, it is breathtaking to see that two prion strains with different characteristics derived from different species contain roughly the same fibril structures (at least half of which is extremely similar). There may be something truly remarkable about this fold that could teach us something about why prions have such unique properties within the realm of amyloids.

We thank the Reviewer for this valuable point. We have amended the text in the Discussion to put more emphasis on the similarity of RML and 263K fibrils. We hope these changes are acceptable.

8. Lastly, the authors might further improve their model geometry while maintaining a good fit to map by relaxing the model in PyRosetta with density and symmetry restraints.

We are grateful to the Reviewer for this suggestion. We ran the Rosetta relax algorithm on our model with density and symmetry restraints. While the overall model geometry was slightly improved, the model:map fit became worse, with several side chains protruding from the density. Based on these findings we have decided to keep the original model.

Reviewer 2

We are very grateful to the Reviewer for their accurate summary of the manuscript and their supportive comments. In particular, for their kind remark stating, *"The paper is very well written and the data are clearly presented."*

The Reviewer had two points for us to address. We have revised the text accordingly and feel that the manuscript has been improved by making these changes.

1. Mass spec indicated that PK N-terminally truncates the rods at 88 with no C-terminal truncation. The cryo-EM data concerns residues 94- 225 – please clarify this apparent discrepancy.

We thank the Reviewer for requesting this clarification. The cryo-EM data concerns residues that are stabilised as part of the amyloid core. Residues 89-93 belong to the protease (PK)-resistant portion of the fibril, but are flexible and therefore not resolved in the cryo-EM density map.

We have now added new text in the Methods and Results describing our mass spectrometry analyses to address the Reviewer's point. We hope these changes are acceptable.

2. Could the authors compare their findings suggesting a PIRIBS structure with previous independent analytical approaches to prion tertiary structures such as NMR, mass spectrometry, and hydrogen-deuterium exchange? Do those findings support the PIRIBS structure?

We thank the Reviewer for this suggestion. There are several previous studies whose data are compatible with the PIRIBS architecture of *ex vivo* prion fibrils. These studies with *ex vivo* material can now be reinterpreted based upon the RML and 263K cryo-EM structures. In fact, a comprehensive review and comment on the historical body of data was done by Caughey and colleagues (BioRxiv and Mol Cell, 2021). Accordingly we have now added a sentence in the Discussion to this effect and hope that this change is acceptable.

REVIEWERS' COMMENTS

Reviewer #1 (Remarks to the Author):

The revised manuscript by Manka et al. is considerably improved and suitably addresses all initial concerns.

The added results and updated form of the discussion present a more comprehensive view of the many subtleties at play when evaluating infectious prion folds.

Reviewer #2 (Remarks to the Author):

I am satisfied with the authors' responses and by their revisions.

2.7 Å cryo-EM structure of ex vivo RML prion fibrils
Nature Communications manuscript NCOMMS-22-00142-A

Point-by-point response to Reviewers.

REVIEWERS' COMMENTS

Reviewer #1 (Remarks to the Author):

The revised manuscript by Manka et al. is considerably improved and suitably addresses all initial concerns.

The added results and updated form of the discussion present a more comprehensive view of the many subtleties at play when evaluating infectious prion folds.

Reviewer #2 (Remarks to the Author):

I am satisfied with the authors' responses and by their revisions.

We are very grateful to the Reviewers for their evaluation of our revised manuscript. They had no further points for us to address.